# YOLO-CID: Improved YOLOv7 for X-ray Contraband Image Detection

**Ning Gan** † , **Fang Wan** † , **Guangbo Lei** *, **Li Xu, Chengzhi Xu, Ying Xiong and Wen Zhou**

School of Computer Science, Hubei University of Technology, Wuhan 430068, China;
102101054@hbut.edu.cn (N.G.); 20021026@hbut.edu.cn (F.W.); 20040038@hbut.edu.cn (L.X.);
xcz911@hbut.edu.cn (C.X.); 19980052@hbut.edu.cn (Y.X.); zw_mmwh@hbut.edu.cn (W.Z.)
* Correspondence: 20000012@hbut.edu.cn
† These authors contributed equally to this work.

**Abstract:** Currently, X-ray inspection systems may produce false detections due to factors such as the varying sizes of contraband images, complex backgrounds, and blurred edges. To address this issue, we propose the YOLO-CID method for contraband image detection. Firstly, we designed the MP-OD module in the backbone network to enhance the model's ability to extract key information from complex background images. Secondly, at the neck of the network, we designed a simplified version of BiFPN to add cross-scale connection lines in the feature fusion structure, to preserve deeper semantic information and enhance the network's ability to represent objects in low-contrast or occlusion situations. Finally, we added a new object detection layer to improve the model's accuracy in detecting small objects in dense environments. Experimental results on the PIDray public dataset show that the average accuracy rate of the YOLO-CID algorithm is 82.7% and the recall rate is 81.2%, which are 4.9% and 3.2% higher than the YOLOv7 algorithm, respectively. At the same time, the mAP on the CLCXray dataset reached 80.2%. Additionally, it can achieve a real-time detection speed of 40 frames per second and 43 frames per second in real scenes. These results demonstrate the effectiveness of the YOLO-CID algorithm in X-ray contraband detection.

**Keywords:** contraband detection; X-ray images; YOLOv7; BiFPN; object detection





## 1. Introduction

In contemporary society, with the diversification of transportation modes and the reduction of travel costs, the density of human traffic in public places is gradually increasing. Therefore, it becomes more and more important to protect people's personal safety and property security in public places. X-rays have the qualities of high energy, a short wavelength, and the ability to penetrate substances, which make them widely used in the fields of video surveillance [1], drone cruising [2], image security inspection [3], etc. At present, security work mainly relies on X-rays to identify contraband such as knives, firearms, and flammable goods, but this identification method mainly relies on the analysis and judgment of the security inspector, so there is greater subjectivity, even among experienced professionals, in the face of a constant stream of X-ray images. This will also produce visual fatigue and thus, in the processing of complex scenes, to the phenomenon of missed or mis-inspection. Therefore, in order to ensure the maximum possible safety of individuals and accelerate the detection efficiency, it is necessary to devise an intelligent detection algorithm with high accuracy and timeliness to identify contraband. However, the poor recognition of objects in the X-ray imaging process, susceptibility to the imaging environment, and high noise levels pose considerable challenges to the construction of X-ray detection models.

There are several traditional methods for the detection of targets in infrared images, including threshold detection [4], the Hough transform [5], and wavelet detection [6]. However, the sensitivity of these methods is influenced by thermal emissivity, making

them vulnerable to interference from the specimen's surface and background radiation. Traditional X-ray detection methods also have two main drawbacks. Firstly, the resulting images have complex structures, poor resolutions, and weak anti-interference abilities and are easily damaged. This makes it difficult to determine the target's shape, size, and location using traditional methods, resulting in low detection accuracy. Secondly, the imaging speed is slow and cannot meet practical demands.

In recent years, object detection algorithms based on deep learning have been widely used in various industries. The YOLO family, a family of regression-based single-stage algorithms, has played an important role in X-ray object detection. [7] YOLO's regression method eliminates the need for complex frameworks, thus reducing the detection time. However, the YOLO algorithm struggles to perform optimally in complex backgrounds where objects overlap and occlude each other. In addition, the problem of multiple color overlaps caused by objects made of different materials when exposed to X-rays needs to be addressed.

YOLOv7 is the latest and most advanced object detection tool in the YOLO series. Its exceptional performance has made it one of the leading real-time object detection methods. Additionally, it has applications in fields such as healthcare, national defense, and security [8–10]. It uses the scalable and efficient layer aggregation network E-ELAN to accelerate model convergence. Rep [11] (RepVGG Block) reparameterization is used to achieve the best trade-off between speed and accuracy during training. Label assignment and auxiliary training heads improve the performance of object detectors in multi-task training. These advantages enable the model to ensure good accuracy and timeliness when detecting X-ray images. However, when directly applied to the X-ray suspected contraband detection field, the YOLO algorithm may encounter some problems:

1. Compared with common scenes, most targets in X-ray images are placed arbitrarily and have directional characteristics. However, the YOLOv7 network's positioning of key information is relatively vague, making it easy to lose key feature information about the directionality of the target. This further increases the difficulty of contraband detection.
2. The objects in X-ray images form a complex background due to overlapping and occlusion. However, there is no corresponding attention mechanism to deal with this complex background, resulting in the inaccurate detection of contraband under such conditions.
3. Although the PAFPN structure in the feature fusion module can enhance the network's representation ability, it does not make full use of the feature map output of each node and does not take into account the different fusion capabilities of each module for features. In response to these challenges, this article targets improvements on the basis of YOLOv7.

This paper proposes an X-ray contraband detection algorithm, YOLO-CID, based on an improved version of YOLOv7 for use in complex scenes. Experiments demonstrate that, in the challenging environment of contraband identification with complex backgrounds, the algorithm can achieve high levels of detection speed and accuracy.

The main contributions of this paper are as follows.

1. This paper proposes the YOLO-CID algorithm for X-ray contraband detection. We conducted ablation and comparative experiments of YOLO-CID on the PIDray [12] dataset and CLCXray [13] dataset. The experimental results show that, compared with current mainstream algorithms, our algorithm has significantly improved detection accuracy and speeds.
2. We implemented a robust new architecture and an enhanced MP-OD model, which builds upon and extends the original MPConv model. We added skip connections between the models and completed the second part (ODConv [14]). This results in a more accurate model with less redundant feature information, greater resilience against background X-ray images, and a faster feature localization speed.
3. We designed the P3-BiFPN module by replacing the original model's PAFPN [15] network with a BIFPN [16] network while retaining the P3 feature fusion layer to preserve shallow semantic information. This improves the network's reasonable application of path resources.

4.   We introduced the shuffle attention mechanism [17], an efficient spatial channel dual attention mechanism, in the neck to improve the network's focus on tiny features.

## 2. Related Works

### 2.1. Traditional Machine Learning Methods for Contraband Detection in X-ray Images

In early machine learning studies of X-ray detection using single-view correlation detection, Turcsany et al. [18] proposed a visual bag-of-words model based on SVM and SURF features. They used starter visual words obtained from clustering to identify contraband in X-ray images, demonstrating the effectiveness of large and distinctly characterized datasets. Riffo et al. also achieved good results by designing an implicit shape model (ISM) for single-view contraband recognition [19]. Kundegorski et al. conducted extensive experiments on X-ray image classification and detection tasks using traditional manual features [20]. By combining multiple manual features, they demonstrated the effectiveness of traditional manual features in X-ray image detection tasks.

Later, multi-view detection techniques were developed to improve the object detection performance by compensating for the incomplete information of single-view imaging. Franzel et al. introduced multi-view imaging for rotating objects and combined SVM with gradient histograms in sliding window detection to improve detection [21]. Bastan et al.conducted a comprehensive evaluation of standard local features for image classification and target detection using the visual bag-of-words model [22]. They extended these features to obtain additional useful information from X-ray images, improving the detection performance.

### 2.2. Deep Learning for Contraband X-ray Image Detection

In recent years, deep-learning-based target detection algorithms have been rapidly developed and have played an important role in X-ray contraband detection, significantly improving the detection accuracy and efficiency compared to traditional algorithms. Mery et al. provided the GDXrays dataset, which contains 8150 X-ray luggage images with guns, hand swords, and blades. The images in the GDXray dataset are grayscale maps with clear target outlines, simple backgrounds, and low object overlap and occlusion [23]. Miao et al. (2019) introduced the larger SIXray dataset, with over 1 million X-ray images containing six types of targets: guns, knives, wrenches, pliers, scissors, and hammers. The SIXray dataset has 8929 labeled images containing targets and a high degree of randomness in target object stacking [24]. Zhao et al. (2022) published the CLCXray dataset to address the overlapping problem in X-ray security images [13]. This dataset has a large amount of data with overlapping phenomena and more accurate annotations compared to previous datasets. The paper also proposes a label-aware mechanism with an attention mechanism that adjusts the feature map according to label information to distinguish different objects in overlapping regions at the high-dimensional feature layer. These large, publicly available datasets provide stable data support for deep learning experiments in this domain and motivate continued development and progress.

In 2012, Krizhevsky et al. proposed the AlexNet network, which achieved excellent results in image classification and demonstrated the potential of deep learning in image processing [25]. Following the success of AlexNet, various classification networks, such as VGG [26], GoogleNet [27], and ResNet [28], YOLOX[29], YOLOv5 [30], and YOLOv, were developed, continuously improving deep learning's classification performance. Akcay [31] et al. applied the AlexNet network to X-ray luggage classification using transfer learning and achieved excellent detection performance compared to traditional machine learning methods. Mery et al. conducted experiments on the GDXray dataset, comparing X-ray luggage classification using bag-of-words models, sparse representation, deep learning, and classical pattern recognition schemes [32]. The results showed that both AlexNet and GoogleNet achieved high recognition rates, indicating the feasibility of using deep learning to design automatic contraband recognition devices. Xu et al. used an attention mechanism to quickly locate unlabeled information in weakly supervised environments where image

information labels were missing [33]. Liu et al. proposed the Faster R-CNN object detection framework based on deep convolutional neural networks (DCNNs) to address detection failures caused by complex image backgrounds [34]. Li et al. improved the YOLOv5 model by compressing channels, optimizing parameters, and proposing a new YOLO-FIRI model for infrared target detection problems such as low recognition rates and high false alarm rates due to long distances, weak energy, and low resolutions [35]. Xiang et al. integrated both MCA and SCA modules into the YOLOx framework, enabling the acquisition of material information for contraband while expanding the model's receptive field, thereby enhancing the detection efficiency [36]. These improvements have significantly impacted the detection quality of contraband detection algorithms. However, real-world contraband detection still faces challenges such as varying item scales and complex backgrounds.

To address existing issues and leverage the unique characteristics of X-ray contraband images, this paper introduces improvements to the MPConv module and feature pyramid module of the YOLOv7 network. Additionally, we incorporate a shuffle attention mechanism and propose the YOLOv7-based YOLO-CID network model. Through ablation and comparative experiments, we demonstrate that the YOLO-CID model is more effective and practical than current mainstream methods and has significant value in the field of X-ray security.

## 3. YOLO-CID

### 3.1. Network Architecture

YOLOv7 is the most advanced object detector in the YOLO series. Its high accuracy and real-time performance have garnered widespread recognition in the field of object detection. In light of this, we propose the YOLO-CID algorithm for X-ray contraband detection, which is based on YOLOv7.

The structure of the YOLO-CID model is shown in the figure below. The model consists of three components: an efficient full-dimensional feature extraction network (MP-OD), an improved bidirectionally weighted feature pyramid network (P3-BiFPN) for feature fusion, and a neck component combined with a shuffle attention mechanism.

In Figure 1, the input image is resized to a uniform size of 640 × 640 pixels to meet the format requirements of the entire network. The resized images are then fed into the backbone network, where the BConv convolutional layer extracts image features at different scales. The MP-OD convolutional layer adopts a parallel strategy to learn the four-dimensional complementary attention of the input channel, output channel, kernel space, and number of kernels without disrupting the original gradient path. This process quickly locates effective features in the model feature map and improves its feature extraction ability. The neck part uses an improved weighted bidirectional feature pyramid, BiFPN-P3. The red line represents our improvement on the original PAFPN. We use the P3 layer, which is the top layer of the neck E-ELEAN module and the MPConv module. The node is deleted, and the root node and end node of the P3 and P4 layers are connected simultaneously. Through a top-down and bottom-up model structure, semantic information of different scales is transferred from shallow to deep layers, outputting three-layer fusion feature maps of different scales. The SA mechanism redistributes the weights in the fused feature map to suppress irrelevant features while enhancing contraband features for more robust representations. Finally, four detection layers at the prediction end predict the confidence, category, and anchor box of the result to obtain the final detection outcome.

Compared to the original YOLOv7 network, this network has shown significant improvements in detection accuracy, speed, and model parameters.

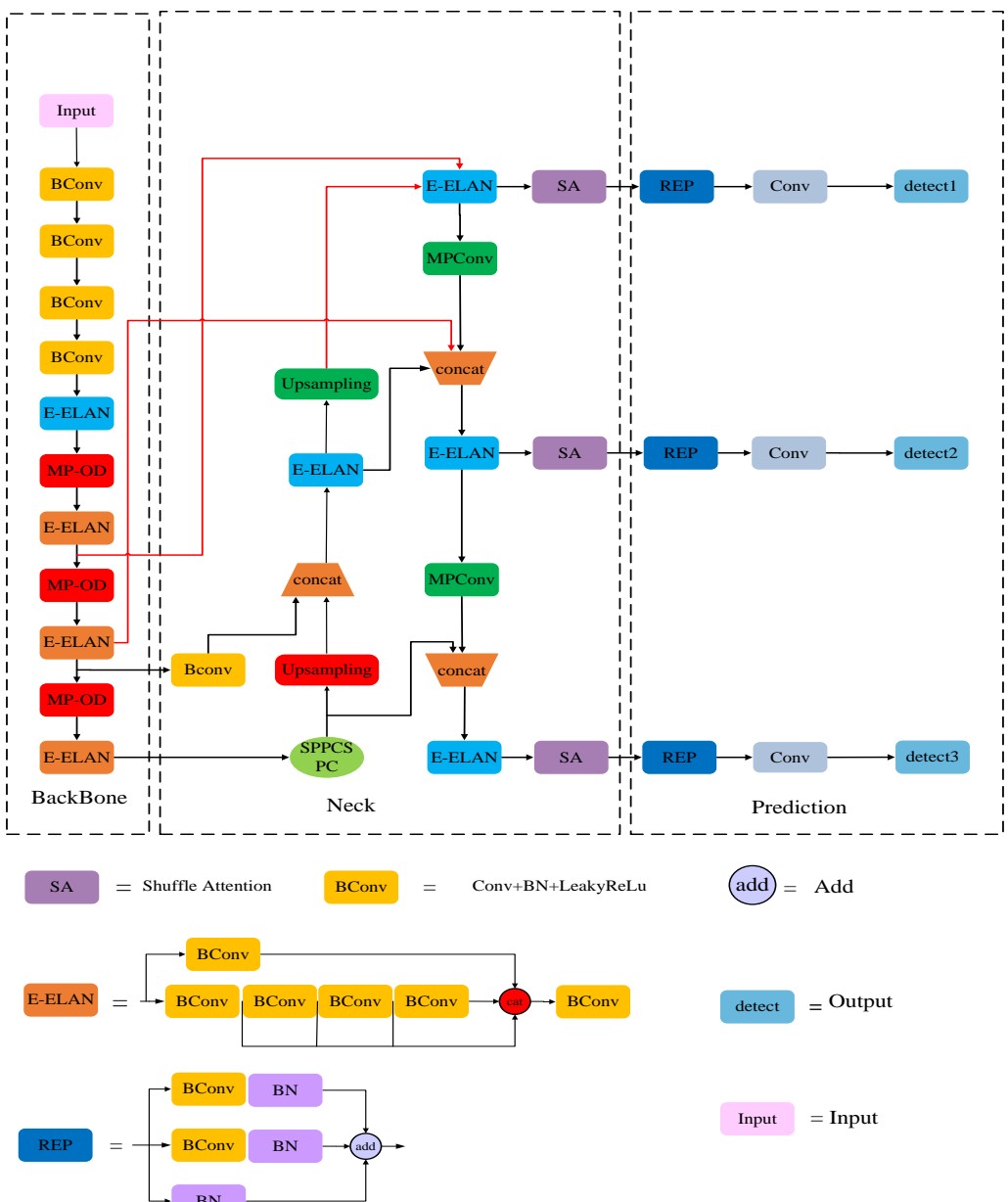

**Figure 1.** YOLO-CID network architecture.

### 3.2. MP-OD Module

The diversity of items in contraband recognition images and their random and variable stacking positions pose a major challenge for the network to effectively extract feature information. To maximize the extraction of key features for X-ray dangerous goods image detection, it is necessary to increase the parameters, depth, and number of channels of the network. However, this leads to increased computational complexity and a larger model size, making deployment more difficult. In the field of contraband identification, it is essential to control the number of model parameters and the amount of computation to ensure timely detection. To solve this problem, we improved the MPconv module of the backbone network and created the OD-MP module, enabling the YOLOv7 model to locate valid features in images more quickly. This improves the timeliness of feature extraction and enhances the object detection performance in complex situations.

In Figure 2, we replace the convolution (CBS) module of the lower branch of the central module with a full-dimensional dynamic convolution (ODConv) module. This allows the model to increase its complexity without increasing the network depth or width,

reducing resource waste. We added a skip connection to the lower branch. When the network generates gradient dispersion due to the introduction of the ODConv module, it can independently select an appropriate path during the backpropagation of the gradient, avoiding branches that produce gradient dispersion. This makes the network fitting more stable and rapid. The specific ODConv structure diagram is shown in Figure 3.

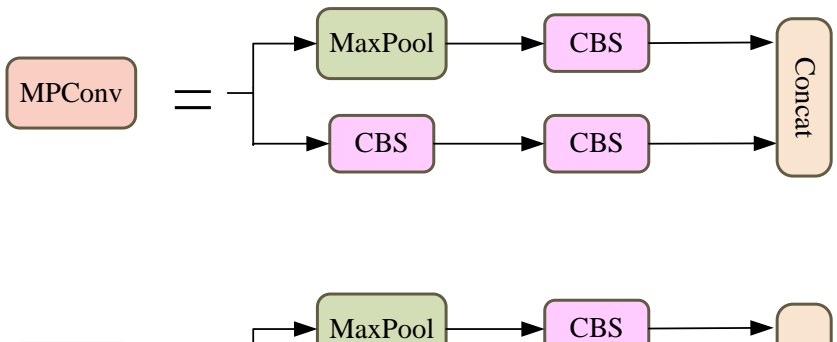

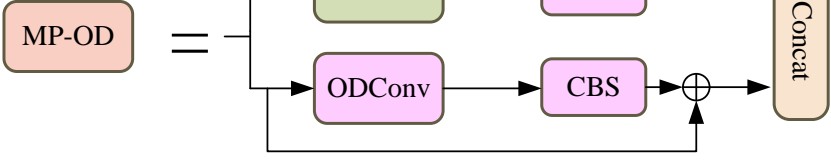

**Figure 2.** Structural comparison of MPConv and MP-OD modules.

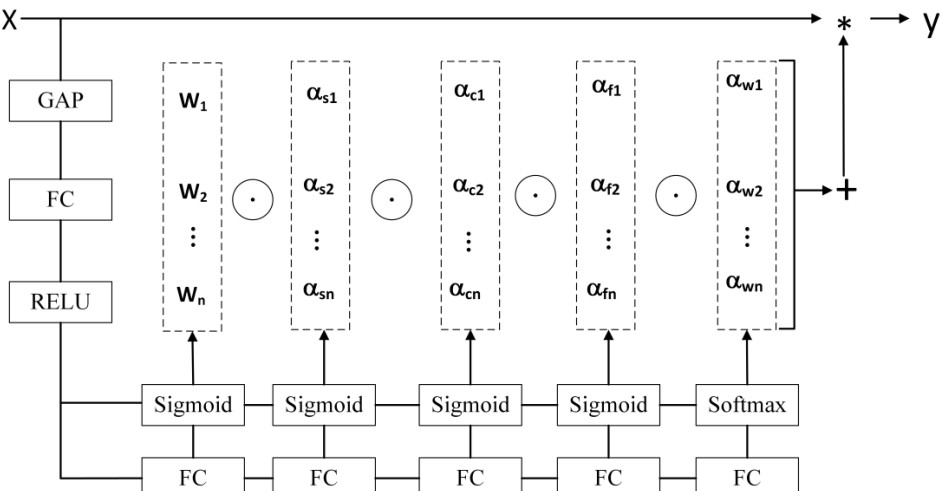

**Figure 3.** ODConv module.

In the convolution kernel $W_i$, $\alpha_{w_i}$ represents the attention scalar, while $\alpha_{s_i}$, $\alpha_{c_i}$, and $\alpha_{f_i}$ represent the attention weights along the spatial dimension, input channel dimension, and output channel dimension, respectively. The input feature vector $X$ has a uniform length through GAP. As shown in the figure, ODConv compresses $X$ into a feature vector of the input channel length through channel average pooling GAP. The feature vector is then mapped to a low-dimensional space through the fully connected layer (FC). After being activated by the ReLU function, it is divided into four head branches. The sigmoid or SoftMax function normalizes it to generate four different types of attention values: $\alpha_{w_i}$, $\alpha_{s_i}$, $\alpha_{c_i}$, and $\alpha_{f_i}$. Its working principle is shown in Formula (1).

$$Z_n = \alpha_{w_n} \odot \alpha_{f_n} \odot \alpha_{c_n} \odot \alpha_{s_n} \odot W_n \tag{1}$$

$$Z_n = \sum_{i=1}^{n} Z_t * X \tag{2}$$

In the equation, $Z_n$ represents the final weight obtained by multiplication in each of the four dimensions of the dynamic convolution kernel. The input feature vector $X$ is length-unified by GAP.

Unlike conditional parameter convolution (CondConv) [37] and dynamic filter convolution (DynamicConv) [38], which only focus on the weight ratio of a single dimension, ODConv uses SE's multi-head attention module to emphasize the importance of the spatial dimension, input channel dimension, and output channel dimension of the convolution kernel space for feature extraction. This module multiplies different attentions along the dimensions of the position, channel, filter, and kernel by progressively multiplying the convolution, providing better performance in capturing rich contextual information. As a result, ODConv greatly improves the feature extraction ability of convolution. More importantly, ODConv achieves better performance with fewer convolution kernels than CondConv and DyConv. Its high-efficiency and lightweight features enable the model to improve its perception of direction, position, and channel information without sacrificing accuracy or incurring a significant computational overhead.

### 3.3. BiFPN-P3 Module

Due to the varying scales of targets to be detected in images, a feature pyramid model (FPN) is commonly used in the feature fusion process of target detection to improve the situation wherein key information from small target objects is ignored during deep convolution. This approach utilizes hierarchical semantic information for feature fusion. The pixel aggregation network (PAFPN) used in the YOLOv7 model adds a low-dimensional to high-dimensional network layer on top of the FPN and transfers semantic information of different scales from shallow to deep. This enriches the semantic information transfer without affecting the location information of the fused feature map, enhancing the network integration effect. However, PAFPN does not fuse the original feature information, resulting in the partial loss of this information and affecting the model's detection accuracy. To address this issue, this paper introduces a bidirectional feature pyramid network (BiFPN) network based on the neck part of the original model. This is a weighted bidirectional (top-down and bottom-up) feature pyramid network, as shown in Figure 4b.

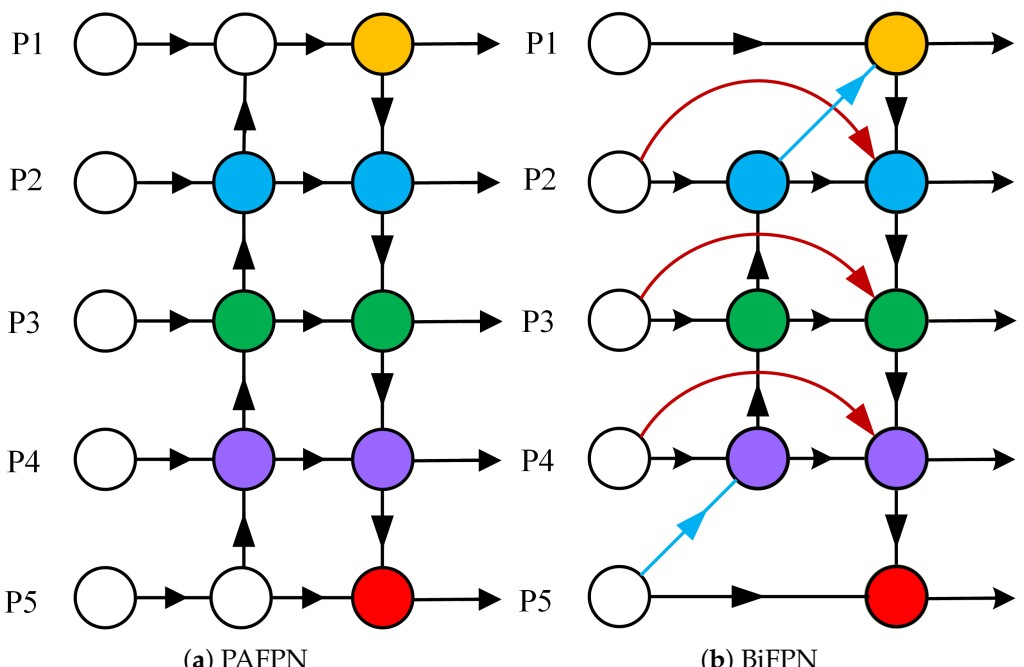

**Figure 4.** Structural comparison of PANFPN and BiFPN modules.

Compared to PAFPN, BiFPN can enhance network feature fusion through a simple residual operation by adding a residual link to the original feature. This strengthens the

network's representation ability. At the same time, BiFPN recognizes that input feature maps of different scales have varying contributions to the network. Therefore, single-input edge nodes that contain less information and have lower contributions to feature fusion are removed from the PAFPN network. This reduces the computational overhead and allows for the better adjustment of each scale feature map's contribution by increasing the weight value after fusion, thereby improving the network's detection speed.

Similar to traditional target detection networks, the feature fusion layer of the YOLOv7 original network is its third layer. Although the BiFPN network adds a new fusion step to the original features and carefully optimizes the network structure to enhance its feature fusion and representation capabilities, the actual detection process of prohibited objects is affected by complex environments. The chaotic placement of contraband and small target objects that are easily obstructed by obstacles during the shooting process can result in the low efficiency of feature fusion and false or missed detections. This paper improves upon the BiFPN network and proposes the BiFPN-P3 model to enhance its ability to locate high-quality features, accelerate the flow of semantic information at different scales, and improve the detection accuracy.

In Figure 5, we retain the feature fusion layer of P3 in the original BiFPN network to preserve its shallow semantic information. Although this approach resulted in a slight increase in computational cost, the improved network architecture enhanced the attention to key information during feature fusion. This made the model more suitable for detecting contraband in complex scenarios.

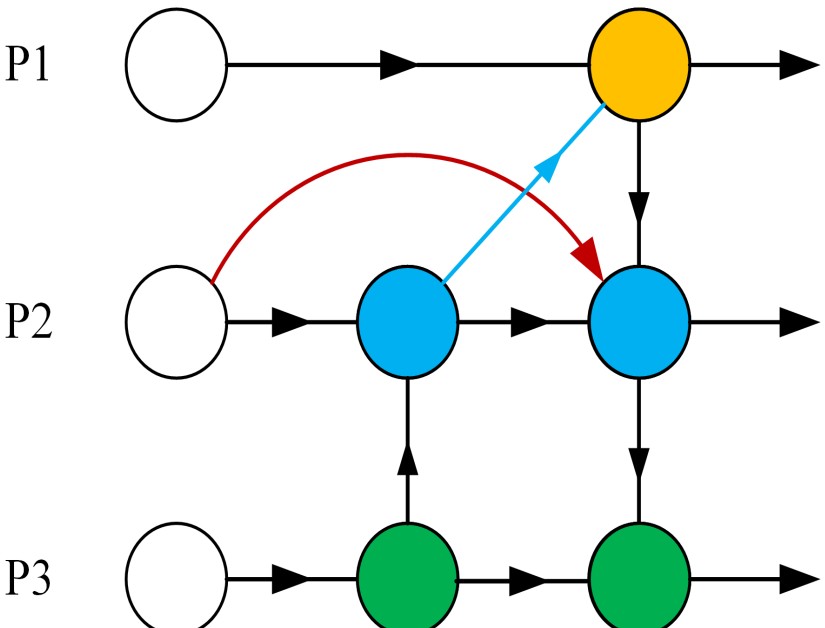

**Figure 5.** The BiFPN-P3 module.

### 3.4. SA Module

Channel attention and spatial attention are used to capture the dependency relationships between image channels and the pixel-level relationships in space, respectively. The SA module efficiently combines these two attention mechanisms without increasing the computational requirements. By adding the SA module to the neck module of YOLOv7, the efficient spatial channel dual attention mechanism (SA) can be fused simultaneously to effectively improve the model's detection performance. As shown in Figure 6, the SA module first groups image channel feature maps to obtain grouped sub-feature maps. The shuffle unit [39] is then used to apply the channel attention mechanism and spatial attention mechanism to each sub-feature map to extract features and capture feature map dependencies. Finally, the channel shuffle operation is used to fuse the summarized feature

maps, establish information communication between sub-feature maps, and use the fused feature maps as the output of the SA module.

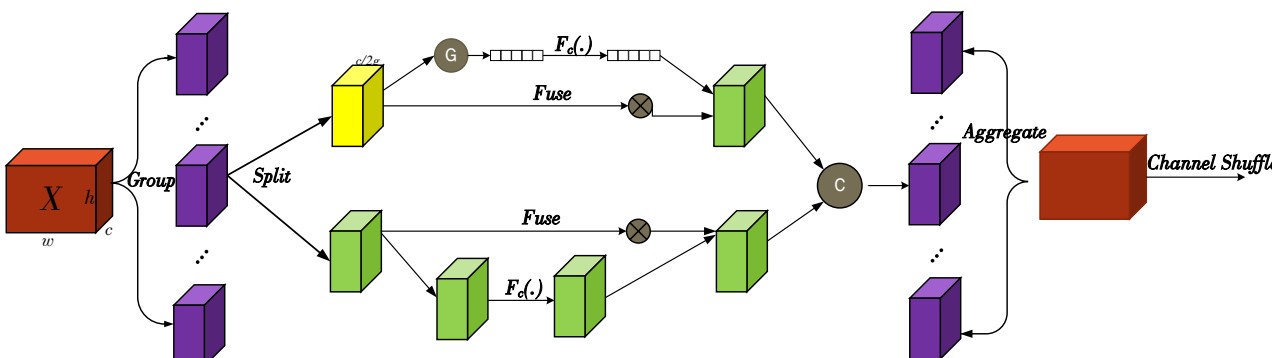

**Figure 6.** Shuffle attention mechanism structure.

The processing steps of the SA mechanism in the network are divided into the following three steps.

1.  Feature grouping: The feature map $S \in R^{CHW}$ of a given length, width, and channel number $W$, $H$, and $C$ is divided into $G$ groups along the channel dimension, denoted as $X = [X_1,\dots,X_G]$, $X_k \in R^{CHW}$. Each sub-feature $X_k$ will gradually capture specific semantic information with training. This part corresponds to the section marked as Group on the leftmost side of the figure above.

2.  Attention mixing: The generated feature $X_k$ is divided into two branches along the channel dimension. The two sub-features are denoted as $X_{k1}$, $X_{k2} \in R^{CHW}$, as shown in the section marked as Split in the middle of the figure. During the processing of feature $X_{k1}$, a group normalization operation is used to accelerate convergence and avoid excessive differences in the values of different features, which can lead to confusion in the learning of lower layer networks. The representation of the enhanced input is then transformed through $F_{c(\cdot)}$. The specific formula is as follows:

$$X'_{k1} = \sigma(W_1 GN(X_{k1}) + b_1) X_{k1} \tag{3}$$

In the equation, GN represents group normalization; $W_1$ and $b_1$ denote the scaling and shifting of the processed feature map. The enhanced feature representation is obtained through the sigmoid activation function.

For feature $X_{k2}$, the channel attention mechanism is employed. To reduce the complexity of the module and improve the processing efficiency, a fast and effective single-layer transformation mode consisting of global average pooling (GAP), scaling, and sigmoid activation is utilized for feature processing. First, channel statistics are generated through GAP to produce channel-level statistics. The specific formula is as follows:

$$s = F_{gp}(X_{k2}) = \frac{1}{H \times W} \sum_{i=1}^{H} \sum_{j=1}^{W} X_{k2}(i,j) \tag{4}$$

In the equation, $\frac{1}{H \times W}\sum_{i=1}^{H}\sum_{j=1}^{W}X_{k2}(i,j)$ denotes the contraction calculation of $X_{k2}$ along the spatial dimension $HW$. The generated $S$ is then screened to obtain the final feature map $X'_{k2}$. The specific formula is as follows:

$$X'_{k2} = \sigma(W_2 \cdot (s) + b2) \cdot X_{k2} \tag{5}$$

Finally, the results of the two types of attention are combined through a concatenation layer to obtain $X'_k = [X'_{k1}, X'_{k2}]$.

3.  Feature aggregation: Similar to ShuffleNetv2, a channel shuffle operation is employed to aggregate all features and facilitate cross-group information exchange along the channel dimension, resulting in the final output feature map.

The aforementioned operations on the feature maps effectively integrate semantic and spatial information across different scales. The terminal attention mechanism improves the model's focus and enhances its detection efficiency in complex scenes.

## 4. Experimental Results and Analysis

### 4.1. Dataset

In order to verify the practicality and effectiveness of YOLO-CID in the field of X-ray contraband detection, we used two public datasets: PIDray and CLCXray. PIDray is a large-scale X-ray benchmark dataset for real-world contraband item detection, covering the detection of prohibited items in various situations, especially intentionally hidden items. The dataset contains more than 47,000 images of prohibited items in 12 categories with pixel-level annotations, including high-quality annotated segmentation masks and bounding boxes. It is currently the largest prohibited item detection dataset. The distribution of each class is shown in Figure 7. The test set is divided into three subsets, easy, hard, and hidden, with the hidden test set focusing on detecting contraband intentionally hidden in clutter. We used the hidden test set as our experiment's test set and divided the PIDray dataset into a training set and a test set at a ratio of 8:2.

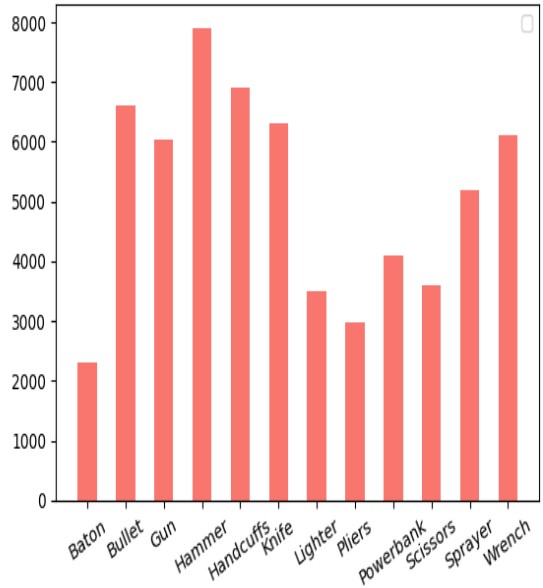

**Figure 7.** Class distribution of the PIDray dataset. The red bar represents the number of each class in the PIDray dataset.

CLCXray was jointly constructed by Tongji University, Beijing University of Posts and Telecommunications, and the University of the Chinese Academy of Sciences. It contains 9565 X-ray security images in 12 categories, including five types of knives (blades, daggers, knives, scissors, Swiss Army knives) and seven types of liquid containers (cans, beverage cartons, glass bottles, plastic bottles, vacuum cups, spray cans, tin cans). The distribution of each class is shown in Figure 8. In our experiment, we used 6696 images as the training set and 2869 images as the test set. Our partitioning of the modified dataset was consistent with that of PIDray.

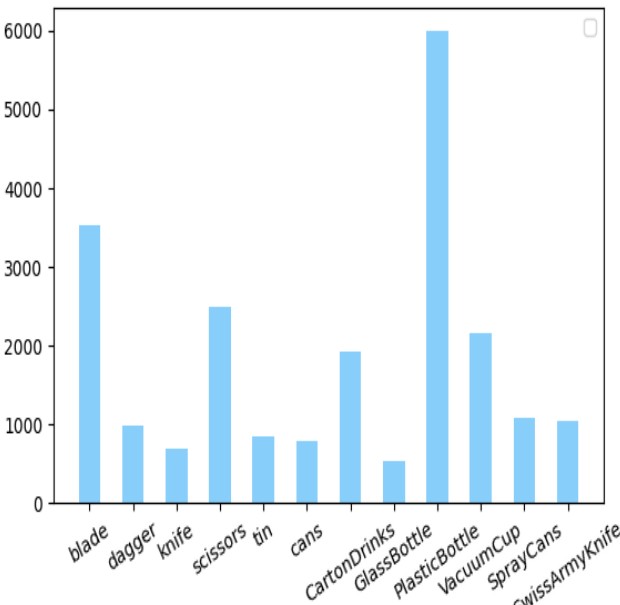

**Figure 8.** Class distribution of the CLCXray dataset. The blue bar represents the number of each class in the CLCXray dataset.

We demonstrated the superiority of the YOLO-CID algorithm through ablation and comparative experiments on the PIDray dataset and CLCXray dataset. The experiments were conducted on a Windows 10 64-bit operating system with an Intel i7-9700k processor and GeForce GTX3060 GPU. The acceleration environment was CUDA 11.6, the deep learning framework was Pytorch 1.12.1, and the programming language was Python 3.7.17. The experimental parameter settings are presented in Table 1.

**Table 1.** Configuration parameters of the experimental platform.

| Parameters | Settings |
|---|---|
| Weights | Yolov7.pt |
| Epochs | 300 |
| Batch size | 16 |
| Hyperparameter file | hyp.scratch.p5.yaml |

### 4.2. Analysis of Ablation Experiments

Three improvements were proposed for the original YOLOv7 algorithm. To verify the value of the proposed modules, ablation experiments were designed by gradually adding the improved modules. The model was trained and tested; '√' indicates the use of this modular approach. The results are shown in Table 2.

**Table 2.** Experimental results of MCS algorithm ablation on the test set of the PIDray dataset and CLCXray dataset.

| Group | MP-OD | BiFPN-P3 | SA | mAP (%) | | $F_1$ Score (%) | |
|---|---|---|---|---|---|---|---|
| | | | | PIDray | CLCXray | PIDray | CLCXray |
| G1 | | | | 64.2 | 75.2 | 72.7 | 78.5 |
| G2 | √ | | | 66.1 | 77.8 | 73.4 | 80.4 |
| G3 | √ | √ | | 69.3 | 78.7 | 75.3 | 81.9 |
| G4 | √ | √ | √ | 70.3 | 80.2 | 77.4 | 82.5 |

The results show that all three improvement points of the YOLO-CID algorithm improved the model's detection performance. In scheme 1, the MP-OD module was used in the backbone network to improve the model's positioning rate. Compared with the original model, the mAP increased by 1.9% and 2.6%, and the F1 score increased by 0.6% and 1.9%. In scheme 2, the PAFPN network of the original model was modified. The results show that the mAP increased by 5.1% and 3.5%, and the F1 score increased by 2.6% and 3.4%. Finally, the shuffle attention mechanism was introduced to increase the detection accuracy by 6.1% and 5.0%, and the F1 score increased by 4.7% and 4.0%. These three changes effectively increased the network's accuracy in identifying contraband.

### 4.3. Algorithm Performance Analysis

According to the experimental results of scheme 1 and scheme 4, under the same conditions, the evaluation index of YOLO-CID exceeded that of the original YOLOv7 algorithm. The mAP50 values on the PIDray and CLCXray datasets reached 70.3% and 80.2%, respectively. The YOLO-CID algorithm significantly improves the detection ability of contraband in complex situations and effectively addresses the issues of missed and false detections in X-ray object detection.

Figure 9 compares the detection accuracy of each category between YOLO-CID and the original YOLOv7 algorithm on the PIDray dataset. As shown, the detection accuracy of our proposed algorithm is higher than that of the original YOLOv7 for all categories. In particular, the detection of lighters, sprayers, and knives has been significantly improved compared to the original model.

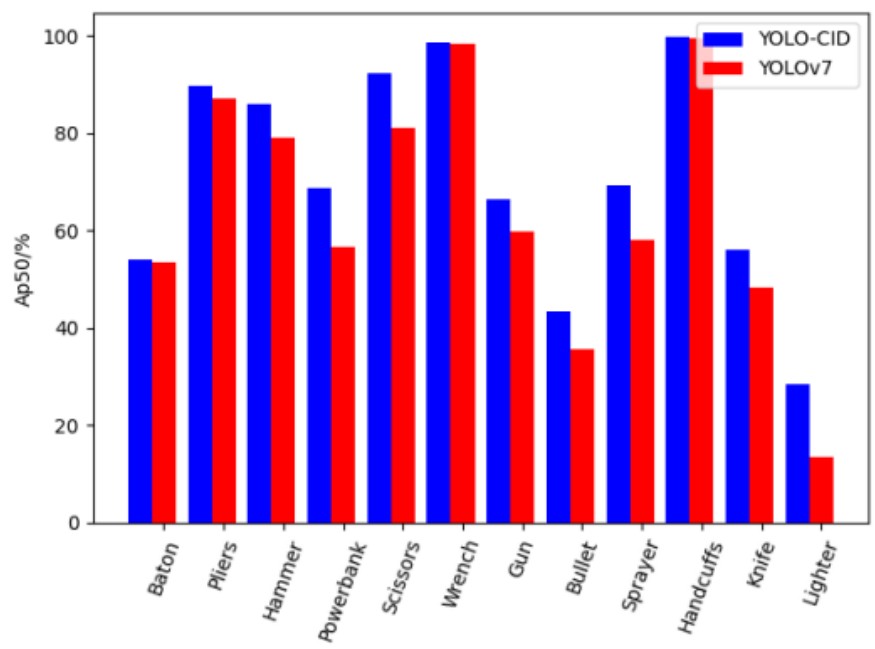

**Figure 9.** Single-class average precision comparison.

Figure 10 presents the confusion matrix for the PIDray dataset using the YOLOv7 model, while Figure 11 displays the confusion matrix for the same test set using the YOLO-CID model. A comparison of the two figures reveals that the detection accuracy for each class has been significantly improved with the YOLO-CID algorithm relative to the original algorithm. This suggests that the YOLO-CID model places greater emphasis on feature information and exhibits superior performance.

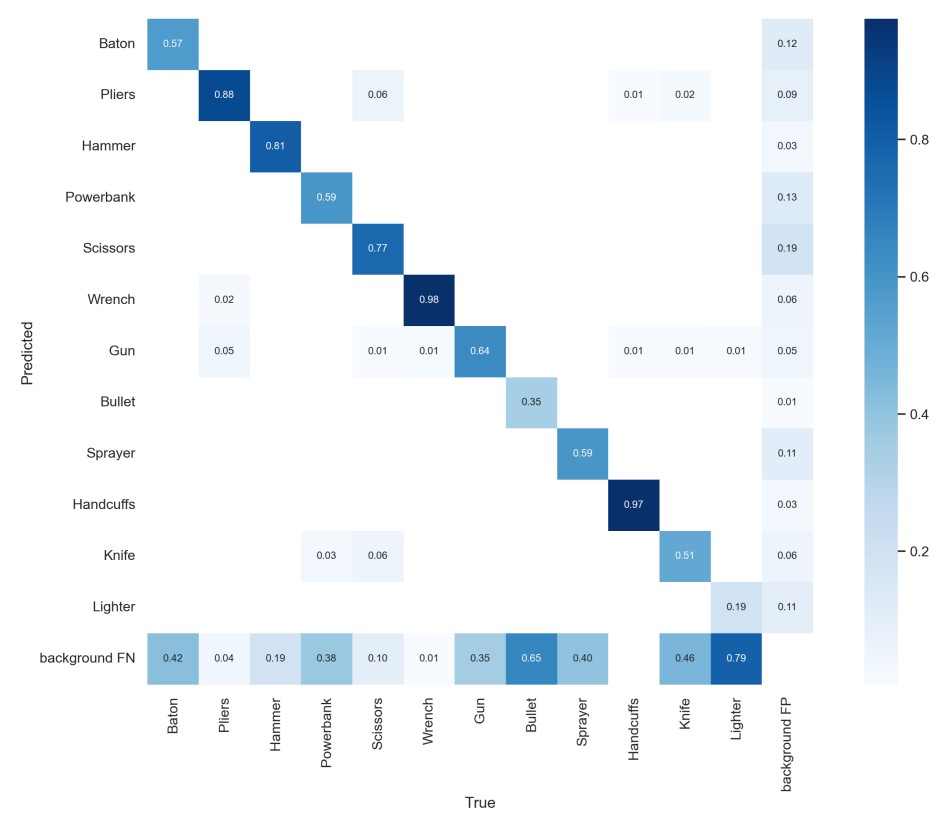

**Figure 10.** Confusion matrix for YOLOv7 network model.

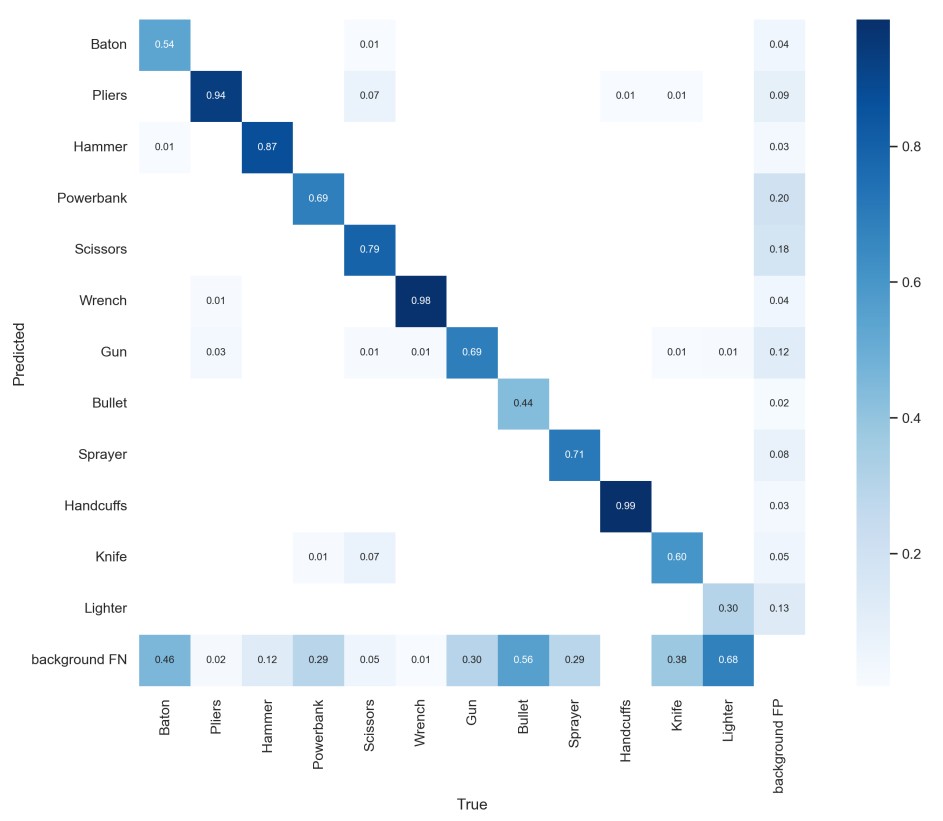

**Figure 11.** Confusion matrix for YOLO-CID network model.

Figure 12 presents the detection results for the YOLO-CID network model and the original YOLOv7 network model on the hidden test set, while Figure 13 presents the detection results for the YOLO-CID network model and the original YOLOv7 network model on the CLCXray test set. It can be seen that YOLO-CID exhibits stronger adaptability and generalization ability in detecting X-ray contraband under simulated real conditions. Compared to the YOLOv7 algorithm, YOLO-CID displays a higher level of confidence when detecting the same object. Additionally, the YOLO-CID algorithm has greatly improved the issues of missed and false detections in contraband detection, demonstrating its superiority and practicality.

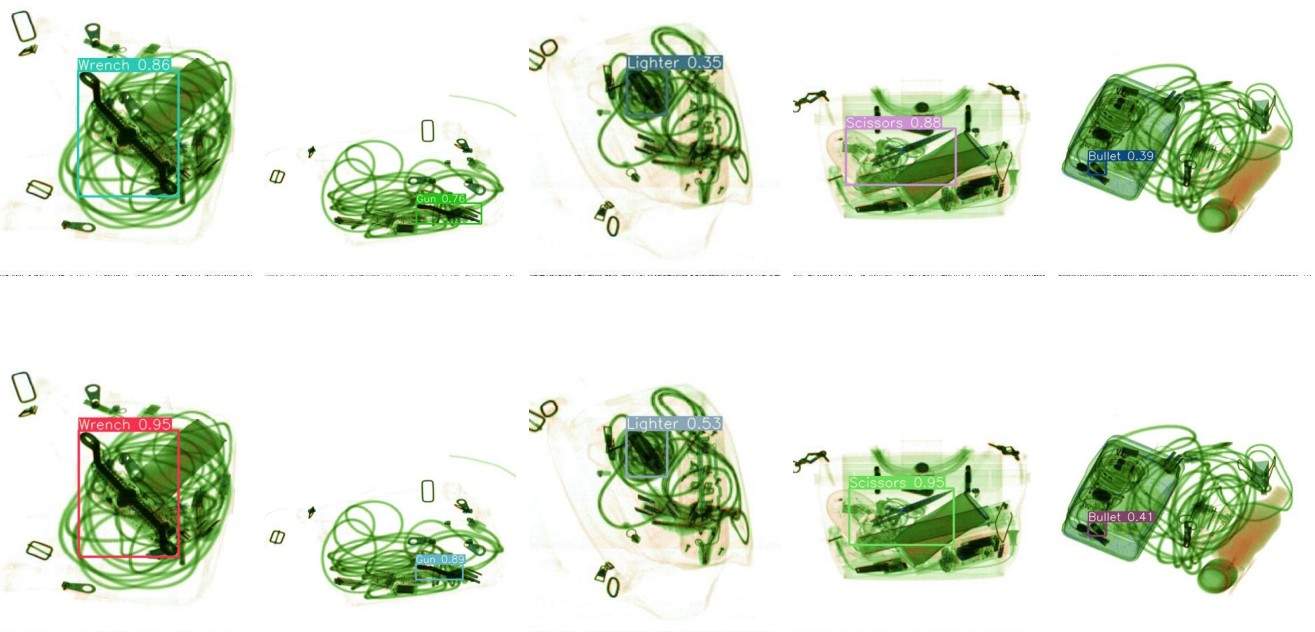

**Figure 12.** Some examples of the detection result on the test set of the PIDray dataset. The first row is the result of YOLOv7, and the second row is the result of YOLO-CID. We used the same four images to compare the performance of the detection models.

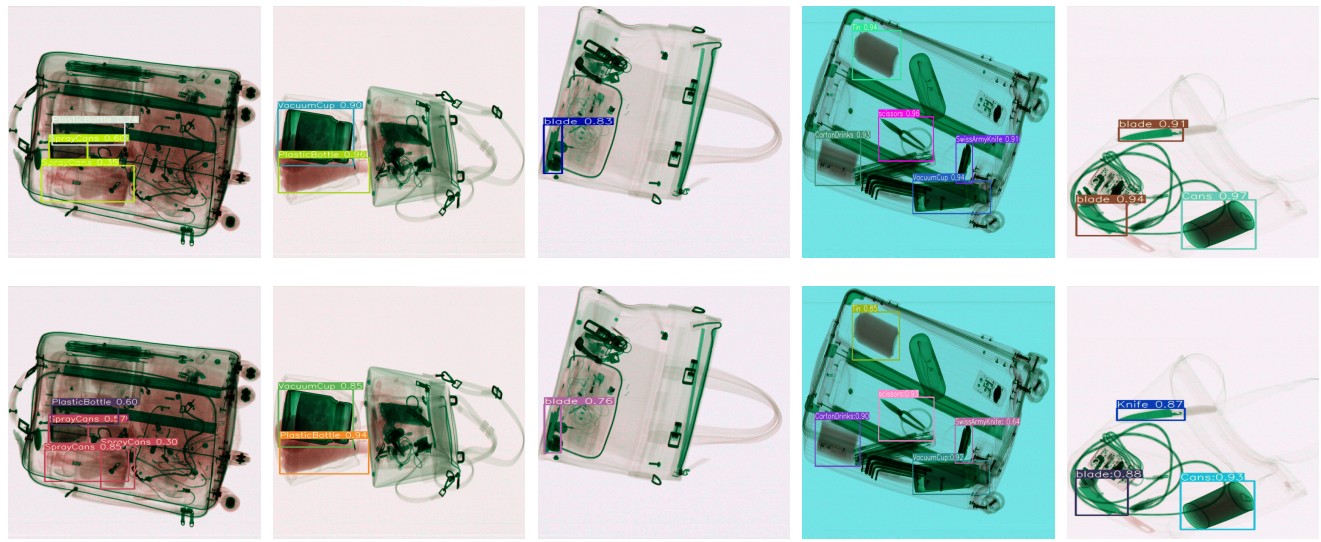

**Figure 13.** Some examples of the detection result on the test set of the CLCXray dataset. The first row is the result of YOLOv7, and the second row is the result of YOLO-CID. We used the same four images to compare the performance of the detection models.

### 4.4. Comparative Experimental Analysis

Table 3 shows the experimental results of different algorithm models on the test set of the PIDray dataset, while Table 4 presents the results on the test set of the CLCXray dataset. The average accuracy of the YOLO-CID algorithm is 70.3% and 80.2%, which are 6.1% and 5.0% higher than in the case of YOLOv7, respectively. The real-time detection speed is 40 frames per second and 43 frames per second, respectively. These results demonstrate that the YOLO-CID algorithm outperforms both single-stage and two-stage algorithms, exhibiting high detection accuracy while meeting the requirements of real-time detection.

**Table 3.** Experimental results comparing different algorithmic models on the PIDray dataset test set.

| Model | AP50 (50%) | FPS |
|---|---|---|
| Faster R-CNN [40] | 42.1 | 13.9 |
| SSD512 [41] | 43.8 | 16.1 |
| YOLOv3 [42] | 69.0 | 34.9 |
| YOLOv5s [30] | 65.5 | 39.2 |
| YOLOv7 | 64.2 | 39.0 |
| Ours | 70.3 | 40.6 |

**Table 4.** Experimental results comparing different algorithmic models on the CLCXray dataset test set.

| Model | AP50 (50%) | FPS |
|---|---|---|
| Cascade R-CNN [43] | 71.4 | 18.0 |
| SSD512 [41] | 66.4 | 21.6 |
| YOLOv3 [42] | 67.2 | 36.7 |
| YOLOv6s [44] | 71.2 | 39.9 |
| YOLOv7 | 75.2 | 41.2 |
| Ours | 80.2 | 43.3 |

Figure 14 illustrates the convergence of the loss functions for various models on the PIDray dataset. As depicted, the bounding box loss of the YOLO-CID algorithm decreased more rapidly during training and it exhibited lower loss values compared to other algorithms. Additionally, its mean average precision (mAP) value was higher. These results demonstrate that the improved algorithm converges more quickly and exhibits a higher degree of alignment between predicted and ground truth frames, thereby proving its effectiveness and superiority.

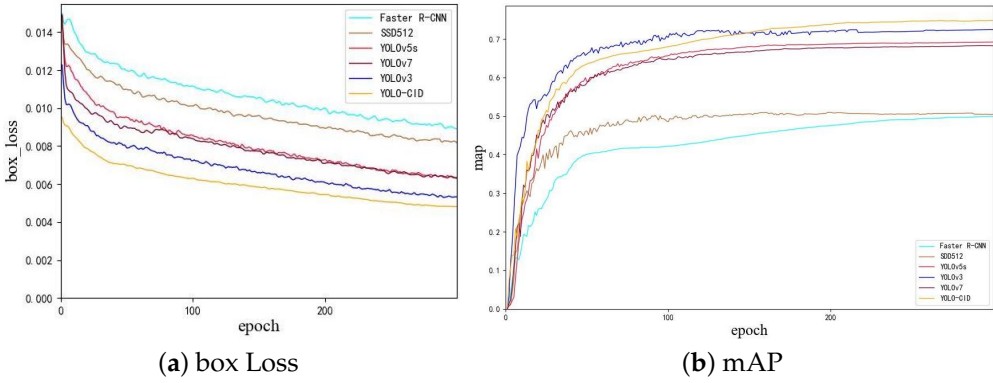

(**a**) box Loss                                                                (**b**) mAP

**Figure 14.** Comparison of evaluation indexes under different models: (**a**) bounding box loss curve, (**b**) map curve.

## 5. Conclusions

To perform real-time X-ray contraband detection, we improved the original YOLOv7 network. We designed the MP-OD module in the backbone of YOLOv7 to enhance the timeliness of feature extraction, optimize the convolutional layer structure of the network, improve the model's ability to extract key information from complex background images, and reduce resource waste. In the neck component, we replaced the path aggregation network of the original model with a simplified version of BiFPN-P3, a bidirectional weighted feature pyramid network, and removed single-input edge nodes containing less PAN information to reduce the computational overhead. We also added an SA mechanism to enhance the model's attention to effective feature information without increasing the computational complexity. Ablation experiments on the extended PIDray and CLCXray datasets showed that these strategies effectively improved the timeliness and detection accuracy in complex background scenes. Comparative experiments with other classic object detection algorithms showed that under the same conditions, our improved YOLOv7 model achieved the highest F1 score and AP value and had a faster detection speed than the other five algorithms, demonstrating its effectiveness for real-time contraband detection.

**Author Contributions:** Conceptualization, N.G. and F.W.; methodology, N.G.; software, N.G. and G.L.; validation, N.G.; formal analysis, L.X. and C.X.; investigation, F.W. and Y.X.; resources, W.Z. and L.X.; data curation, N.G. and G.L.; writing—original draft preparation, N.G.; writing—review and editing, F.W. and G.L.; visualization, N.G.; supervision, C.X. and Y.X.; project administration, W.Z.; funding acquisition, W.Z. and L.X. All authors have read and agreed to the published version of the manuscript.

**Funding:** This research was funded by the National Natural Science Foundation of China (Grant No. 62202147) and the Science and Technology Research Project of the Education Department of Hubei Province (Grant No. B2021070).

**Data Availability Statement:** Not applicable.

**Conflicts of Interest:** The authors declare no conflict of interest.

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
