# Peer review of "YOLO-CID: Improved YOLOv7 for X-ray Contraband Image Detection"

_electronics, doi:10.3390/electronics12173636_

Round 1

Reviewer 1 Report

This paper presents YOLO-CID, an improved-variant of YOLOv7 for contraband image detection in X-ray images. The results showed that the variant performs well on 1 X-ray image dataset. Despite the good performance, this paper is of limited novelty and does not present a clear motivation over the variance. Further, the literature review of the paper is far from satisfactory, while the experiments are only conducted on 1 dataset. The authors should consider addressing the following concerns before the paper could be accepted:

1. Why YOLO-CID instead of YOLOv7? Apart from a better performance (which is the equivalence of a better feature obtained), why is YOLO-CID designed in the proposed manner for X-ray images? Are there any particular aspects of the CID task and X-ray images that motivates the design and variance from YOLOv7? The authors should first and foremost present the motivation of the proposed method clearly and precisely. Note that a mere better performance is a week motivation since there are much more ways (such as the use of larger models, LLMs, or some training and evaluation engineering skills and tricks) if the purpose is merely to improve performance.

2. The two core variants: Omni-dimensional Dynamic Convolution (ODConv) and Bidirectional Feature Pyramid Network (BiFPN), are NOT novel techniques, and have already been used in various works and task. (BiFPN is NOT cited properly). The novelty of this work is therefore limited (at the very least, or NONE to be frank) as its just YOLOv7 + ODConv + BiFPN, which it could work on CID and the X-ray images.

3. There are quite a number of X-ray CID works that are not discussed in this work, a few are listed here:

a. Yinsheng Zhang, Wenxiao Xu, Shanshan Yang, Yongjie Xu, and Xinyuan Yu, "Improved YOLOX detection algorithm for contraband in X-ray images," Appl. Opt. 61, 6297-6310 (2022)

b. B. Song, R. Li, X. Pan, X. Liu and Y. Xu, "Improved YOLOv5 Detection Algorithm of Contraband in X-ray Security Inspection Image," 2022 5th International Conference on Pattern Recognition and Artificial Intelligence (PRAI), Chengdu, China, 2022, pp. 169-174, doi: 10.1109/PRAI55851.2022.9904110.

c. Chen, Hao, and Zhe-Ming Lu. "Contraband detection based on deep learning." Journal of Information Hiding and Multimedia Signal Processing 13, no. 3 (2022): 165-177.

d. C. Liqun and L. Wanxin, "Faster R-CNN-Based Method for Detecting Contraband Targets in X-ray Images," 2023 IEEE 3rd International Conference on Electronic Technology, Communication and Information (ICETCI), Changchun, China, 2023, pp. 1577-1582, doi: 10.1109/ICETCI57876.2023.10176625.

4. The figures are not imformative or properly displayed. For Figure 1, where is the BiFPN part? For Figure 3, it is not properly displayed (height value too large). For Figures 4 and 5, what does the nodes stand for? What do the different colors mean? What do the arrows stand for?

Overall the paper is readable and can be followed through. There are some typos in this manuscript and the authors are advised to go through them carefully.

Reviewer 2 Report

1. The language of the presentation is complex enough to understand the modifications of the neural network model structure. A more detailed description of what these modifications give is desirable;

2. The names of paragraphs 4.2 and 4.3 on page 10 match. At the same time, the last paragraph of paragraph 4.2 and the only paragraph of paragraph 4.3 are completely identical.

3. It is not shown how the trained model works on data that is not included in the dataset used for training and testing;

4. Issues related to balancing training data have not been considered.

Round 2

Reviewer 1 Report

The authors have addressed my concerns in the previous review with more concise motivation description, better formatting, and more proper citation. I thank the authors for their effort.

There are still minor typos, please go through the paper.
